# Sequence-to-Set Generative Models

**Longtao Tang**[1], **Ying Zhou**[2] **and Yu Yang**[*1,3]

[1]School of Data Science, City University of Hong Kong, Hong Kong, China
[2]Department of Economics and Finance, City University of Hong Kong, Hong Kong, China
[3]Hong Kong Institute for Data Science, City University of Hong Kong, Hong Kong, China
`longttang2-c@my.cityu.edu.hk`, `{ying.zhou, yuyang}@cityu.edu.hk`

## Abstract

In this paper, we propose a sequence-to-set method that can transform any sequence generative model based on maximum likelihood to a set generative model where we can evaluate the utility/probability of any set. An efficient importance sampling algorithm is devised to tackle the computational challenge of learning our sequence-to-set model. We present GRU2Set, which is an instance of our sequence-to-set method and employs the famous GRU model as the sequence generative model. To further obtain permutation invariant representation of sets, we devise the SetNN model which is also an instance of the sequence-to-set model. A direct application of our models is to learn an order/set distribution from a collection of e-commerce orders, which is an essential step in many important operational decisions such as inventory arrangement for fast delivery. Based on the intuition that small-sized sets are usually easier to learn than large sets, we propose a size-bias trick that can help learn better set distributions with respect to the $\ell_1$-distance evaluation metric. Two e-commerce order datasets, TMALL and HKTVMALL, are used to conduct extensive experiments to show the effectiveness of our models. The experimental results demonstrate that our models can learn better set/order distributions from order data than the baselines. Moreover, no matter what model we use, applying the size-bias trick can always improve the quality of the set distribution learned from data.

## 1 Introduction

Learning a generative model [Goodfellow et al., 2014, Kingma and Welling, 2014] from data to generate random samples is a fundamental unsupervised learning task. In many real applications, the data, such as customer orders and drug packages, can be represented by sets. Learning a set generative model can boost many important applications such as simulating users' demand of e-commerce orders for inventory arrangement [Wu et al., 2021]. However, in literature, little attention has been paid to generative models for sets.

In this paper, we study the problem of learning a set generative model from data. Specifically, we study sets containing some categorical items from a finite ground set. Some previous studies [Benson et al., 2018, Wu et al., 2021] proposed traditional statistical models to learn the set generative model, while how to employ deep learning to design set generative models remains largely untouched. Meanwhile, deep generative models for sequence data [Hochreiter and Schmidhuber, 1997, Schäfer and Zimmermann, 2006, Cho et al., 2014, Devlin et al., 2018] have been extensively studied. Since the only difference between a sequence and a set is that the order of items does not matter in a set, we propose to leverage a sequence generative model to build a set generative model.

---

[*]Yu Yang is the corresponding author.

A natural idea of converting a random sequence to a random set is to just ignore the order of items. However, for a set, the number of possible sequences that can be regarded as equivalent to the set is exponential to the number of items. This brings us a serious computational challenge where we need to enumerate all the equivalent sequences for a set when learning the parameters of the model. Traditional statistical models [Benson et al., 2018, Wu et al., 2021] along this line either make unrealistic independent assumptions on items or run a learning algorithm of complexity exponential to set sizes to only deal with small sets. These constrained treatments undermine the model's ability to capture the complex correlations among items in sets.

To tackle the computational challenge while still well capturing the correlations among items, in this paper, we make the following technical contributions.

In Section 3, we first show that we can convert any sequence generative model based on maximum likelihood, such as RNN [Schäfer and Zimmermann, 2006] and DeepDiffuse [Islam et al., 2018], to a set generative model. We call our method Sequence-to-Set. To address the computational challenge, we propose an efficient importance sampling algorithm to estimate the gradient of the log-likelihood function of our Sequence-to-Set model.

In Section 4, to capture the complex item correlations, we propose two deep models GRU2Set and SetNN, which are instances of our Sequence-to-Set model. GRU2Set directly uses the GRU as an essential building block, while SetNN is designed for obtaining permutation invariant set embeddings when learning a set generative model. We discuss that GRU2Set and SetNN can express any set distributions as long as we increase the model capacity.

To learn a better set distribution from data, in Section 5, we reveal that we often need to modify the empirical size distribution of the training data and propose a heuristic to do so. This size-bias trick can be applied to all set generative models aiming at learning a set distribution.

We conduct extensive experiments using two e-commerce datasets TMALL and HKTVMALL, and report the experimental results in Section 6. The experimental results clearly show the superiority of our models to the baselines and the effectiveness of our size-bias trick.

## 2 Related Work

Learning a set generative model is often regarded as learning a choice model for subset selection. Along this line, Benson et al. [2018] proposed a discrete choice model (DCM) to model the utility/probability that users select a specific set. Wu et al. [2021] utilized representation learning to design a parameterized Markov Random Walk to model how users select random sets from an item graph. Both [Benson et al., 2018] and [Wu et al., 2021] are set generative models, though they are not deep models. To achieve practical computational efficiency, both [Benson et al., 2018] and [Wu et al., 2021] only handle small sets and compromise on modeling high-order item correlations.

To model the permutation invariant nature of sets, Ruiz et al. [2017] used the sum or max of item embedding as the set embedding. Zaheer et al. [2017] systematically investigated the theory of permutation invariant functions. Stelzner et al. [2020] proposed a generative model for point clouds based on GAN [Goodfellow et al., 2014] and Set Transformer [Lee et al., 2019]. Although a point cloud is a set of 2D-coordinates, we cannot use the point cloud generative model [Stelzner et al., 2020] to solve our problem as the model outputs continuous random variables instead of categorical data. Kosiorek et al. [2020] presented another generative model for sets of real-valued vectors based on Set Transformer [Lee et al., 2019] and their generating process is based on VAE [Kingma and Welling, 2014]. A disadvantage of using GAN or VAE to train a generative model is that it would be difficult to evaluate the probability density of a random sample. Normalizing Flows are powerful generative models based on solid statistical theories. However, the study of Discrete Flows is still in its infancy [Hoogeboom et al., 2019, Tran et al., 2019] and no existing works have investigated using Discrete Flows to learn set distributions.

Multi-label classification also needs to model the permutation invariance of sets of labels. Yang et al. [2019] applied reinforcement learning with a permutation-invariant reward function to remove the impact of label orders. Yazici et al. [2020] proposed to align the labels of a sample with its predicted labels before calculating the loss of the sample. However, our task in this paper is an unsupervised learning task. We cannot apply the loss functions in [Yang et al., 2019, Yazici et al., 2020] since we do not have "labels" (or feature vectors if one regards a set as a set of labels) for training samples.

To sum up, existing works on set generative models either deal with sets of real-valued vectors which are different from categorical sets, or adopt non-deep models which need to compromise on capturing item correlations to make the computation efficient. It is still urgent to design deep generative models for categorical sets and propose efficient algorithms to deal with computational challenges.

# 3 Sequence-to-Set Generative Models

In this section, we introduce how to convert a sequence generative model to a set generative model and how to optimize the model parameters.

## 3.1 Converting Sequences to Sets

We first characterize the sequence generative model that we want to convert to a set generative model.

**Definition 1 (Sequence Generative Model)** *A sequence generative model is a model with a sequential generating process, which can be represented by a sequence of states and actions* $[s_0, a_0, s_1, a_1, ..., s_{T-1}, a_{T-1}, s_T]$, *where $s_i$ is the state at time $i$, $a_i$ is the action taken at time $i$ and $s_T$ is the end state. We call $[s_0, a_0, s_1, a_1, ..., s_{T-1}, a_{T-1}, s_T]$ a **generating path**. Like a finite-state machine, the sequence generative model uses each action $a_i$ to **deterministically** update the current state $s_i$ to $s_{i+1}$. For each state $s$, the sequence generative model has a corresponding probability distribution $p(a \mid s; \boldsymbol{\theta})$ indicating the probability of choosing action $a$ when the current state is $s$, where $\boldsymbol{\theta}$ is the parameters of the sequence generative model.*

Our definition of the sequence generative model can cover a wide range of models that can generate items in a sequential manner. For example, the commonly used sequence model RNN is an instance, where the state is the history vector $H$ and an action is to choose the next word/item. Moreover, influence diffusion models such as the Independent Cascade (IC) model [Goldenberg et al., 2001] and the DeepDiffuse model [Islam et al., 2018] are also covered by Definition 1, since in these models the state at time $t$ is the set of activated nodes until time $t$ and the action $a_t$ is to select the group of nodes to be activated at time $t + 1$.

To convert a sequence generative model in Definition 1 to a set generative model, we need to associate each state $s$ with a **state-associated set** $S_s$. For example, in an RNN, we can define $S_{s_t}$ as the set of items generated in the first $t$ steps. In the IC model, we can define $S_{s_t}$ as the set of nodes activated at or before time $t$.

We say that a generating path $l = [s_0, a_0, \ldots, s_{T-1}, a_{T-1}, s_T]$ can **induce** a set $S$ if $S = S_{s_T}$. Let $L(S) = \{l \mid l \text{ can induce } S\}$ be the **path set** of $S$. As the sequence generative model has the probability distribution $p(a \mid s; \boldsymbol{\theta})$ indicating the probability of taking $a$ as the next action given the current state $s$, we can derive the probability of having a path $l = [s_0, a_0, \ldots, s_{T-1}, a_{T-1}, s_T]$ as

$$p(l; \boldsymbol{\theta}) = p(a_0 \mid s_0; \boldsymbol{\theta}) \times p(a_1 \mid s_1; \boldsymbol{\theta}) \times \cdots \times p(a_T \mid s_T; \boldsymbol{\theta}).$$

Therefore, given a sequence generative model with parameters $\boldsymbol{\theta}$, the probability to induce a set $S$ is

$$p(S; \boldsymbol{\theta}) = \sum_{l \in L(S)} p(l; \boldsymbol{\theta}).$$

## 3.2 Parameter Learning

Suppose we have a collection (multiset) of observed sets $\mathcal{S} = \{S_1, S_2, ..., S_N\}$. We want to use $\mathcal{S}$ as the training data to learn a sequence model which can generate the observed sets. We adopt maximum likelihood estimation to learn the parameters $\boldsymbol{\theta}$. Specifically, the loss function is

$$\mathcal{L}(\boldsymbol{\theta}) = -\sum_{i=1}^{N} \log p(S_i; \boldsymbol{\theta}) = -\sum_{i=1}^{N} \log \left( \sum_{l \in L(S_i)} p(l; \boldsymbol{\theta}) \right)$$

We apply stochastic gradient descent based algorithms to minimize the loss function. The stochastic gradient w.r.t. an observation $S$ is $\nabla \log p(S; \boldsymbol{\theta}) = \nabla \log \left( \sum_{l \in L(S)} p(l; \boldsymbol{\theta}) \right)$. As in the log-sum term of $\nabla \log p(S; \boldsymbol{\theta})$, the number of paths that can induce $S$ is often exponential to $|S|$ (in some

sequence generative model the number of such paths is even infinite), it is very challenging to evaluate $\nabla \log p(S; \boldsymbol{\theta})$ exactly.

To tackle the computational challenge in computing $\nabla \log p(S; \boldsymbol{\theta})$, we first find that $\nabla \log p(S; \boldsymbol{\theta})$ can be regarded as an expectation as follows.

$$\nabla \log p(S; \boldsymbol{\theta}) = \frac{\sum_{l \in L(S)} \nabla p(l; \boldsymbol{\theta})}{p(S; \boldsymbol{\theta})} = \frac{\sum_{l \in L(S)} p(l; \boldsymbol{\theta}) \nabla \log p(l; \boldsymbol{\theta})}{p(S; \boldsymbol{\theta})} = E_{l \sim p(l|S;\boldsymbol{\theta})}[\nabla \log p(l; \boldsymbol{\theta})],$$

where $p(l \mid S; \boldsymbol{\theta}) = \frac{p(l;\boldsymbol{\theta})}{p(S;\boldsymbol{\theta})}$ is a posterior distribution indicating the probability that a given set $S$ is generated by the path $l$. Therefore, $\nabla \log p(S; \boldsymbol{\theta})$ can be seen as the expected value of $\nabla p(l; \boldsymbol{\theta})$ where the random variable $l$ follows the posterior distribution $p(l \mid S; \boldsymbol{\theta})$.

We can apply Monte Carlo method to estimate $\nabla \log p(S; \boldsymbol{\theta}) = E_{l \sim p(l|S;\boldsymbol{\theta})}[\nabla \log p(l; \boldsymbol{\theta})]$, where the key point is to sample generating paths from $p(l|S; \boldsymbol{\theta})$. A naive idea is to use rejection sampling, where we use the model with parameters $\boldsymbol{\theta}$ to generate random paths and reject those that cannot induce $S$. Obviously such a naive implementation would be extremely inefficient as the rejection rate would be very high.

To sample $l$ from $p(l|S; \boldsymbol{\theta})$ more efficiently, we devise an important sampling method as follows. Let $\tau(a, s, S)$ be an indicator function, where $\tau(a, s, S) = 1$ if $\exists l = [s_0, a_0, \dots, s_{T-1}, a_{T-1}, s_T]$ such that $l$ induces $S$ and $\exists i \in \{0, 1, \dots, T-1\}$, $s_i = s \wedge a_i = a$. In our sampling algorithm, for a state $s_t$, we only consider actions $a_t$ such that $\tau(a_t, s, S) = 1$. We take such an action $a_t$ with probability

$$q(a_t \mid s; \boldsymbol{\theta}, S) = \frac{p(a_t \mid s; \boldsymbol{\theta})}{\sum_{a:\tau(a,s,S)=1} p(a \mid s; \boldsymbol{\theta})}.$$

Therefore, the proposal distribution $q(l; \boldsymbol{\theta}, S)$ for $l = [s_0, a_0, \dots, s_{T-1}, a_{T-1}, s_T]$ in our importance sampling is

$$q(l; \boldsymbol{\theta}, S) = q(a_0 \mid s_0; \boldsymbol{\theta}, S) \times q(a_1 \mid s_2; \boldsymbol{\theta}, S) \times \cdots \times q(a_{T-1} \mid s_{T-1}; \boldsymbol{\theta}, S)$$

The (unnormalized) importance weight of $l$ is

$$r(l; \boldsymbol{\theta}) = \left( \sum_{a:\tau(a,s_0,S)=1} p(a \mid s_0; \boldsymbol{\theta}) \right) \times \left( \sum_{a:\tau(a,s_1,S)=1} p(a \mid s_1; \boldsymbol{\theta}) \right) \times \cdots \times \left( \sum_{a:\tau(a,s_{T-1},S)=1} p(a \mid s_{T-1}; \boldsymbol{\theta}) \right)$$

Clearly, we have

$$r(l; \boldsymbol{\theta}) q(l; \boldsymbol{\theta}, S) \propto p(a_0|s_0; \boldsymbol{\theta}) p(a_1|s_1; \boldsymbol{\theta}) \dots p(a_{T-1}|s_{T-1}; \boldsymbol{\theta}) = p(l; \boldsymbol{\theta}).$$

This shows the correctness of the sampling method.

**An EM perspective** Using our importance sampling algorithm to estimate $\nabla \log p(S; \boldsymbol{\theta})$ in minimizing the loss function $\mathcal{L}(\boldsymbol{\theta})$ can be interpreted from an Expectation-Maximization algorithm's perspective. Our method is actually a Monte Carlo EM algorithm if we treat a generating path $l$ as a latent variable. In the E-step, we sample the latent variable $l$ from the posterior distribution $p(l|S; \boldsymbol{\theta}^{old})$. Our importance sampling can boost the efficiency of the E-step. In the M-step, we reconstruct an object function which is $\sum_i^M r(l_i) R^{-1} \log p(l_i; \boldsymbol{\theta}^{old})$, where $M$ is the number of samples and $R = \sum_{i=1}^M r(l_i)$. To maximize this function, we update $\boldsymbol{\theta}$ by gradient based methods, such as $\boldsymbol{\theta}^{new} = \boldsymbol{\theta}^{old} + \eta \sum_{i=1}^M r(l_i) R^{-1} \nabla_{\boldsymbol{\theta}} \log p(l_i; \boldsymbol{\theta}^{old})$, where $\eta$ is the learning rate and $\sum_{i=1}^M r(l_i) R^{-1} \nabla_{\boldsymbol{\theta}} \log p(l_i; \boldsymbol{\theta}^{old})$ is estimated by our importance sampling algorithm.

**MRW as an instance of Sequence2Set** We show that the Markov Random Walk (MRW) model [Wu et al., 2021] is also an instance of our Sequence2Set model. The sequence model in MRW is a random walk on the item graph [Wu et al., 2021] where a node is an item and we have a special stop item $\kappa$. The state $s_t$ at time $t$ is associated with a set $S_{s_t}$ which includes all the items visited until time $t$. An action $a_t$ taken at time $t$ given the state $s_t$ is to jump to an item $i_t$ from $i_{t-1}$, the item visited at time $t-1$. Once the stop item $\kappa$ is visited, the random walk terminates. The probability of jumping from $i_{t-1}$ to $i_t = i$ is parameterized as $p(i_t = i \mid s_{t-1}; \boldsymbol{\theta})$, where $\boldsymbol{\theta}$ is the embeddings of items. Specifically, in MRW, $p(i_t = i \mid s_{t-1}; \boldsymbol{\theta}) = \frac{\exp(\boldsymbol{e}_i^\top \boldsymbol{e}_{i_{t-1}})}{\sum_j \exp(\boldsymbol{e}_j^\top \boldsymbol{e}_{i_{t-1}})}$, where $\boldsymbol{e}_i$ is the embedding

vector of item $i$. When an action $a$ is jumping to an item $i \neq \kappa$, the indicator $\tau(a, s, S) = 1$ if $i \in S$. When an action $a$ is jumping to the stop item $\kappa$, $\tau(a, s, S) = 1$ if $S_s = S$. Wu et al. [2021] propose an algorithm of complexity exponential to $|S|$ to calculate the gradient $\nabla \log p(S; \boldsymbol{\theta})$ for learning the item embeddings and thus, only small sets of size at most 4 are considered in [Wu et al., 2021]. If we adopt our important sampling algorithm, we can deal with large sets.

# 4 GRU2Set and SetNN

In this section, we propose two deep Sequence-to-Set models GRU2Set and Set Neural Networks (SetNN). We first introduce some general settings that will be used in both GRU2Set and SetNN.

**Item embedding and stop item** We also adopt item embeddings, which should be learned from the training dataset, to parameterize the probability $p(a \mid s; \boldsymbol{\theta})$ in the sequence generative model. We assign an embedding vector $\boldsymbol{e}_i \in \mathbb{R}^d$ to each item $i$. Similar to MRW [Wu et al., 2021], we also add a stop item $\kappa$ where once we add $\kappa$ to the set associated to the current state, we stop the generating process. The stop item $\kappa$ also has an embedding vector $\boldsymbol{e}_\kappa \in \mathbb{R}^d$.

**Sparse item graph** In our Sequence-to-Set model, the indicator function $\tau(a, s, S)$ is a key for us to efficiently sample generating paths from the posterior distribution $p(l \mid S; \boldsymbol{\theta})$. Intuitively, $\tau(a, s, S)$ indicates what actions are valid given the current state $s$ and the induced set $S$. To further boost the efficiency of our importance sampling algorithm, we make extra constraints on the items that can be added in the next step when we are given the current state $s$. To do so, we build an item graph $G = \langle V, E \rangle$, where $V$ is the set of items plus the stop item $\kappa$ and $(i, j) \in E$ if there is a set $S$ in the training dataset $\mathcal{S}$ such that $i \in S$ and $j \in S$. The stop item is connected to all the other items. When using our importance sampling algorithm, given the current state $s$ associated with a set $S_s \subseteq S$, $\tau(a, s, S) = 1$ if action $a$ is adding an item $i \in S \cap \text{Neighbor}(S_s)$, where $\text{Neighbor}(S_s)$ is the set of neighbors to items in $S_s$. When starting from the first state $s_0$ associated with a set

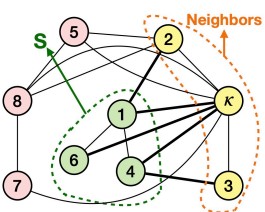

Figure 1: An example of sparse item graph. The item $\kappa$ is the stop item.

$S_{s_0} = \emptyset$, we set $\text{Neighbor}(S_{s_0}) = V \setminus \{\kappa\}$. In real set datasets such as TMALL and HKTVMALL used in our experiments, such an item graph built from training datasets is often very sparse. Therefore, we call the item graph the **sparse item graph**. Clearly, the sparsity of the item graph can help boost the efficiency of generating random sets from a learned model.

**Choice vector** In our model design, the conditional probability $p(a \mid s; \boldsymbol{\theta})$ is the key to control the generating process. Exploiting the idea of representation learning, we represent each state $s$ with an embedding vector $\boldsymbol{c}_s \in \mathbb{R}^d$. We call $\boldsymbol{c}_s$ the **choice vector** of $s$. Given $s$, we control the probability of adding $i$ as the next item as

$$p(i \mid s; \boldsymbol{\theta}) = \frac{\exp(\boldsymbol{c}_s^\top \boldsymbol{e}_i)}{\sum_{j \in \text{Neighbor}(S_s)} \exp(\boldsymbol{c}_s^\top \boldsymbol{e}_j)}.$$

The sparsity of the item graph can make computing the normalization term in $p(i \mid s; \boldsymbol{\theta})$ efficient. Note that in MRW [Wu et al., 2021], the choice vector $\boldsymbol{c}_s$ is the embedding of the last item visited so far. We will see that in GRU2Set and SetNN, we use deep neural nets to aggregate the embeddings of all the items visited so far to obtain the choice vector $\boldsymbol{c}_s$. The initial choice vector $\boldsymbol{c}_{s_0}$ is trainable.

## 4.1 GRU2Set

We leverage GRU [Cho et al., 2014] to build our first sequence-to-set model, since GRU is one of the most popular sequential model based on RNN and its computational cost is low. We set the history vector $H$ of GRU as a $d$-dimensional vector and $H$ is regarded as the choice vector of the current state. The initial history vector $H_0$ is trainable and $H$ is updated according to the design of GRU. The state-associated set $S_{s_t}$ for the state $s_t$ at time $t$ is just the set of items added in the first $t$ steps of the sequence generating process of GRU. When the stop item is added, we terminate the generating process. Due to limited space, the overview of GRU2Set is put in Appendix.

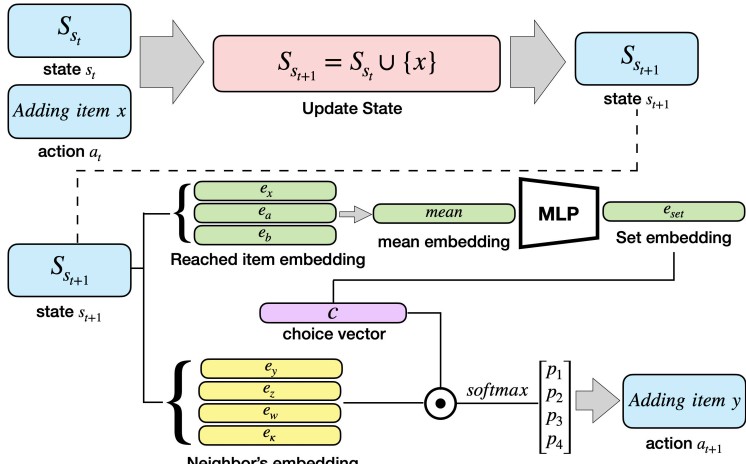

Figure 2: Overview of SetNN. We do not show the situation when we add the stop item. If the action is adding the stop item, we end the process immediately.

## 4.2 SetNN

The choice vector $\boldsymbol{c}_s$ can be regarded as the embedding of the set $S_s$. In GRU2Set, the choice vector, which is the history vector $H$, is sensitive to the order of items added to $S_s$. As sets are permutation invariant, we hope to have a sequence-to-set model which can also produce permutation invariant representations of sets. Therefore, we develop the Set Neural Networks (SetNN) where the key idea is to design a permutation invariant aggregation of embeddings of items in a set. The following theorem by Zaheer et al. [2017] guides our model design.

**Theorem 1** ([Zaheer et al., 2017]) *A function $f(X)$ operating on a set $X$ having elements from a countable universe, is a set function, i.e., invariant to the permutation of instances in $X$, iff it can be decomposed in the form $\rho(\sum_{x \in X} \Phi(x))$ for suitable transformations $\Phi$ and $\rho$.*

$\Phi(x)$ can be viewed as the embedding of $x$ and the sum over $\Phi(x)$ can be replaced by other aggregate operators such as mean or pooling. In SetNN, we use an MLP to express the function $\rho$ in Theorem 1 to obtain the embedding (choice vector) $\boldsymbol{c}_s$ of $S_s$ for any state $s$. Fig 2 shows the overview of SetNN.

## 4.3 Expressive Power of GRU2Set and SetNN

Compared to SetNN, our GRU2Set model uses the history vector $H$. By the expressive power of RNN [Schäfer and Zimmermann, 2006], as we increase the embedding dimension $d$, hidden neural units and layers in GRU module, the history vector $H$ of GRU2Set can approximate any function on a finite history path arbitrarily close. Then, of course, $H$ can express any permutation invariant function on sets which implies that the expressive power of GRU2Set is more than SetNN.

To explore the expressive power of SetNN, we first give a recursive definition of the probability that a set $S$ is generated by SetNN, denoted by $p(S)$, as follows.

$$p(S) = p(\kappa|S)\gamma(S) \quad (\kappa \text{ is the stop item})$$
$$\gamma(S) = \sum_{x \in S} p(x|S \setminus \{x\})\gamma(S \setminus \{x\}) \quad (1)$$
$$\gamma(\emptyset) = 1,$$

where $p(x|S \setminus \{x\})$ is the conditional probability of adding item $x$ to the set $S \setminus \{x\}$ and $\gamma(S)$ is the probability of reaching a state $s$ such that $S_s = S$ in SetNN. The following theorem shows that Eq. (1) is general enough to cover all possible set distributions.

**Theorem 2** *For any distribution $q(S)$ on all subsets of a ground set of items, there exists a group of transition probabilities $p(x|S)$ and $p(\kappa|S)$, such that $q(S) = p(S)$ holds for any $S$, where $p(S)$ is defined in Eq. (1).*

The proof of Theorem 2 can be found in Appendix B. As we increase the capacity of the MLP in SetNN, SetNN can represent any conditional probability $p(x|S)$ and $p(\kappa|S)$, and as a result of Theorem 2, SetNN and GRU2Set can express any set distributions.

# 5   Size-Bias Trick for Improving Learned Set Distributions

In this section, we propose a **size-bias** trick that can help reduce the distance between the ground truth set distribution $p^*(S)$ and the learned distribution $p(S)$, where $p(S)$ can be learned by any generative model not limited to our Sequence-to-Set models. We first decompose the ground truth distribution $p^*(S)$ when $|S| = k$ as

$$p^*(S) = p^{*(k)}(S)p_k^*,$$

where $p_k^* = \sum_{A:|A|=k} p(A)$ is the probability of generating a set of size $k$, and $p^{*(k)}(S) = \frac{p(S)}{p_k^*}$ is the probability that a random size-$k$ set is $S$. Suppose we know the real size distribution $p_k^*$ and we have $q^{(k)}(S)$ as an estimation of $p^{*(k)}(S)$ for each $k$. How can we construct a distribution $q(S)$ close to $p^*(S)$? An intuitive way is to combine $q^{(k)}(S)$ and $p_k^*$ such that $q(S) = q^{(k)}(S)p_k^*$ if $|S| = k$ [Stelzner et al., 2020, Benson et al., 2018]. However, we argue that this intuitive way is not always the optimal plan as follows.

Let $K$ be the largest possible set size. To find the best size distribution $q_k$ to collaborate with our estimation $q^{(k)}(S)$, we consider the following optimization aiming at minimizing the KL-divergence between the distribution $q(S) = q^{(k)}(S)q_k$ and the ground truth $p^*(S)$.

$$
\begin{aligned}
\min_{q_1,\ldots,q_K} \quad & KL(q\|p^*) = \sum_S q(S)\log\frac{q(S)}{p^*(S)} \\
s.t. \quad & q(S) = q_k \times q^{(k)}(S), \; \forall |S| = k \\
& 0 \le q_k \le 1, \forall k \\
& \sum_{k=1}^{K} q_k = 1
\end{aligned}
\tag{2}
$$

Since we do not know $p^*(S)$, we cannot solve Eq. (2) exactly. However, we still can tell if setting $q_k = p_k^*$ is the optimal solution to some extend.

Let $f_k(q_k) = \sum_{S:|S|=k} q(S)\log\frac{q(S)}{p^*(S)}$. We have $KL(q\|p^*) = \sum_k f_k(q_k)$. Note that $f_k(q_k)$ only depends on $q_k$. We can interpret Eq. (2) as a portfolio optimization problem. There are $K$ projects. $q_k$ is the investment for project $k$ and $-f_k(q_k)$ is the revenue of project $k$. To decide the best portfolio $q_k$, we check the derivative of $f_k(q_k)$, which indicates the marginal revenue. We have

$$f'_k(q_k) = 1 + \sum_{S:|S|=k} q^{(k)}(S)\log\frac{q(S)}{p^*(S)}$$

By taking $q_k = p_k^*$, we have

$$f'_k(p_k^*) = 1 + \sum_{S:|S|=k} q^{(k)}(S)\log\frac{q^{(k)}(S)}{p^{*(k)}(S)} = 1 + KL\left(q^{(k)}(S)\|p^{*(k)}(S)\right)$$

As the estimation quality of $q^{(k)}(S)$ often varies for different $k$, the derivative $f'_k(p_k^*)$ probably also varies for different $k$, making $q_k = p_k^*$ not a stationary point. Therefore, we probably need to set $q_k$ different from $p_k^*$ for solving Eq. (2). We first make the following reasonable assumption.

**Assumption 1** *Small-sized sets are easier to learn than large sets. In other words, $q^{(k)}(S)$ estimates $p^{*(k)}(S)$ better as the set size $k$ decreases.*

Based on Assumption 1, compared to $p_k^*$, we should "invest" more on small set sizes to construct the "portfolio" (size distribution) $q_k$. Therefore, we set $q_k$ slightly bigger than $p_k^*$ for small $k$, and set $q_k$ smaller than $p_k^*$ for big $k$. To address the issue of not knowing $p_k^*$, we can resort to the empirical size

Table 1: Basic statistics

| dataset | #item | #training sets | average #testing sets | average #edge |
|---------|-------|----------------|-----------------------|---------------|
| TMALL | 1,363 | 100,000 | 393,226 | 60,956 |
| HKTVMALL | 782 | 200,000 | 616,652 | 53,853 |

Table 2: Size distribution of TMALL and HKTVMALL

| | 1 | 2 | 3 | 4 | 5 | 6 | $\geq 7$ |
|---|---|---|---|---|---|---|---|
| TMALL | 38% | 18% | 14% | 10% | 7% | 5% | 8% |
| HKTVMALL | 42% | 17% | 11% | 8% | 6% | 5% | 12% |

distribution of the training dataset, which is a good estimation of the real size distribution $p_k^*$. We construct a biased size distribution $q_k$ by adopting the following heuristic.

**Constructing Biased Size Distribution** Suppose the empirical size distribution of the training data $\mathcal{S}$ is $[p_1, p_2, ..., p_K]$. We first calculate the rest proportion by $r_k = p_k/(p_k + p_{k+1} + ... + p_K)$, which means $p_i = (1 - r_1)...(1 - r_{i-1})r_i$. Then we calculate $r'_k = \max\{r_k + k\sqrt{|V|/|\mathcal{S}|}, 1\}$, where $V$ is the set of all items. We construct a biased size distribution $[q_1, q_2, \ldots, q_K]$ by setting $q_i = (1 - r'_1)...(1 - r'_{i-1})r'_i$.

## 6 Experiments

**Datasets** We did our empirical study on two real-world datasets of customer orders from online e-commerce platforms. This first dataset is TMALL (`https://www.tmall.com`) which contains **1363** items and orders from August 2018 to March 2019. We treated the collection of orders in each month as an instance and in total we have 8 instances for the TMALL dataset. The second dataset is HKTVMALL (`https://www.hktvmall.com`) which contains **728** items form the supermarket sector and orders from February 2020 to September 2020. Similar to TMALL, we also treated HKTVMALL as 8 instances where each instance is the collection of orders in each month. For both datasets, We split all the orders in a month to a training dataset $\mathcal{S}_{train}$ and a testing dataset $\mathcal{S}_{test}$, where the size of $\mathcal{S}_{train}$ is 100,000 for Tmall and 200,000 for HKTVmall. The datasets can be found in the source code. We report the sparsity of item graphs and statistics of the datasets in Table 1. The set size distribution of each dataset is shown in Table 2.

**Experiment Setting and Evaluation** We treat the method of directly using the histogram of $\mathcal{S}_{train}$ as the **benchmark** method, as it is a natural and unbiased method to estimate the set distribution. Note that this benchmark method has a major disadvantage that its cannot model probabilities of any sets not showing in $\mathcal{S}_{train}$. We also compare our models with two baselines which are DCM (Discrete Choice Model) [Benson et al., 2018] and MRW [Wu et al., 2021].

Following [Wu et al., 2021], for each method (except the benchmark), we generated a collection $\mathcal{S}_{pred}$ of 10,000,000 random sets and used the empirical set distribution of $\mathcal{S}_{pred}$ to approximate the set distribution learned by the model. We regarded the empirical distribution of the testing data $\mathcal{S}_{test}$ as the pseudo ground truth. Since the bigger $\mathcal{S}_{test}$ is, the more accurate the pseudo ground truth is, we set the size of $\mathcal{S}_{test}$ bigger than $\mathcal{S}_{train}$ as shown in Table 1. Denote by $N_{test}(S)$ the number of sets that are $S$ in $\mathcal{S}_{test}$. Let $N_{pred}(S)$ be the number of sets that are $S$ in $\mathcal{S}_{pred}$. We used the $\ell_1$-distance between a model's (approximate) distribution and the pseudo ground truth to evaluate the effectiveness of the model. Specifically, the $\ell_1$-distance can be calculated as follows.

$$\ell_1(\mathcal{S}_{test}, \mathcal{S}_{pred}) = \sum_{S \in \mathcal{S}_{test} \cup \mathcal{S}_{pred}} |\frac{N_{test}(S)}{|\mathcal{S}_{test}|} - \frac{N_{pred}(S)}{|\mathcal{S}_{pred}|}| \tag{3}$$

We set the embedding dimension of all methods as 10. For each training and testing dataset, the biased size distribution was the same for all methods and was obtained by the heuristic introduced at the end of Section 5. For all experiments, the MLP of SetNN only has one hidden layer which contains 50 neural units. All the experiments were ran on a CPU with 10 cores. The optimizer used by us is RMSProp with default parameter in PyTorch. The source code and data used in our experiments can be found at `https://github.com/LongtaoTang/SetLearning`.

Table 3: The performance of each groups on TMALL. G0 stands for the first group of data.

| | G0 | G1 | G2 | G3 | G4 | G5 | G6 | G7 | Average | STDEV |
|---|---|---|---|---|---|---|---|---|---|---|
| Benchmark | 1.11 | 1.02 | 0.91 | 0.90 | 0.94 | 0.94 | 0.93 | 0.85 | 0.95 | 0.07 |
| +Size Bias | 1.06 | 0.97 | 0.87 | 0.86 | 0.90 | 0.90 | **0.89** | 0.81 | 0.91 | 0.07 |
| DCM | 1.17 | 1.08 | 0.95 | 0.94 | 1.00 | 0.99 | 0.99 | 0.89 | 1.00 | 0.08 |
| +Size Bias | 1.10 | 1.01 | 0.90 | 0.89 | 0.94 | 0.93 | 0.93 | 0.84 | 0.94 | 0.07 |
| MRW | 1.14 | 1.07 | 0.99 | 0.96 | 1.02 | 1.02 | 1.03 | 0.92 | 1.02 | 0.06 |
| +Size Bias | 1.06 | 0.97 | 0.88 | 0.87 | 0.90 | 0.92 | 0.95 | 0.84 | 0.92 | 0.06 |
| SetNN | 1.12 | 1.01 | 0.92 | 0.89 | 0.90 | 0.91 | 1.01 | 0.89 | 0.96 | 0.08 |
| +Size Bias | **1.04** | **0.93** | **0.84** | **0.82** | **0.85** | **0.87** | **0.89** | **0.79** | **0.88** | 0.07 |
| GRU2Set | 1.11 | 1.04 | 0.89 | 0.90 | 0.94 | 0.94 | 1.00 | 0.90 | 0.97 | 0.07 |
| +Size Bias | **1.02** | **0.93** | **0.83** | **0.82** | **0.85** | **0.86** | **0.89** | **0.80** | **0.87** | 0.06 |

Table 4: The performance of each groups on HKTVMALL. G0 stands for the first group of data.

| | G0 | G1 | G2 | G3 | G4 | G5 | G6 | G7 | Average | STDEV |
|---|---|---|---|---|---|---|---|---|---|---|
| Benchmark | 0.73 | 0.87 | 0.91 | 0.84 | 0.81 | 0.87 | 0.81 | 0.80 | 0.83 | 0.05 |
| +Size Bias | **0.70** | 0.83 | 0.88 | 0.81 | **0.79** | 0.84 | 0.78 | **0.77** | 0.80 | 0.05 |
| DCM | 0.76 | 0.89 | 0.93 | 0.88 | 0.84 | 0.89 | 0.84 | 0.82 | 0.86 | 0.05 |
| +Size Bias | 0.72 | 0.84 | 0.89 | 0.83 | 0.80 | 0.85 | 0.79 | 0.78 | 0.81 | 0.05 |
| MRW | 0.80 | 0.91 | 0.94 | 0.88 | 0.88 | 0.90 | 0.85 | 0.85 | 0.88 | 0.04 |
| +Size Bias | 0.74 | 0.84 | 0.88 | 0.82 | 0.83 | 0.85 | 0.79 | 0.80 | 0.82 | 0.04 |
| SetNN | 0.82 | 0.86 | 1.00 | 0.84 | 0.89 | 0.92 | 0.84 | 0.84 | 0.88 | 0.05 |
| +Size Bias | **0.71** | **0.82** | **0.87** | **0.79** | **0.79** | **0.83** | **0.76** | 0.78 | **0.79** | 0.04 |
| GRU2Set | 0.83 | 0.87 | 0.94 | 0.84 | 0.93 | 0.94 | 0.86 | 0.82 | 0.88 | 0.05 |
| +Size Bias | **0.70** | **0.80** | **0.85** | **0.78** | **0.79** | **0.82** | **0.75** | **0.75** | **0.78** | 0.04 |

**Experiment Results**   Due to limited space, we only report the main experimental results. More detailed experimental results can be found in Appendix D and E.

Table 3 and Table 4 show the experimental results. We place the results of using the size-bias trick below the original results. We find that both DCM and MRW never beat the benchmark method, no matter the size-bias trick is played or not. A possible reason is that in our experiments we did not make size constraints on sets, while Wu et al. [2021] constrained the set size to be at most 4 and Benson et al. [2018] set the maximum size of sets to be 5. Big sets in our data may have negative effects on learning both DCM and MRW.

We can see that applying the size-bias trick can always reduce the $\ell_1$-distance significantly, which demonstrates the effectiveness of this trick. Before applying the size-bias trick, the benchmark method has the best performance which is slightly better than our models GRU2Set and SetNN. However, after applying the size-bias trick, our GRU2Set model becomes the best and it outperforms the benchmark method by 4% on TMALL and 2.5% on HKTVMALL. SetNN also outperforms the benchmark method after using the size-bias trick. This suggests that GRU2Set and SetNN may learn probabilities of small sets better than the benchmark method.

The SetNN model performs slightly worse than GRU2Set but the gap between them is small. A possible reason is that GRU2Set has stronger expressive power than SetNN as illustrated in Section 4.3. GRU2Set records the order of items added to the set, while SetNN ignores such order to achieve permutation invariant set embeddings. This shows that GRU2Set includes more information in the generating process than SetNN. However, we want to emphasize that SetNN can produce permutation invariant set embeddings while GRU2Set cannot. If we have a downstream task that needs set embeddings, SetNN may have more advantages in this case.

# 7 Conclusion

In this paper, we present a Sequence-to-Set method to build generative models for set data and the SOTA method MRW is an instance of our method. To utilize deep learning in learning set distributions, we further design two models, GRU2Set and SetNN, which are two instances of our Sequence-to-Set model. To learn better set distributions from data, we also propose a size-bias trick. Experimental results on two e-commerce order datasets clearly show that our models outperform the baselines including two SOTA methods in this line of research.

For future work, we will explore using more sophisticated deep learning modules in our model, such as replacing the MLP in SetNN with Set Transformer and substituting the RNN in GRU with more advanced sequence models. For potential negative societal impacts, learning the embeddings of items and sets might cause user privacy leakage as many other embeddings might do. Research on how to hide personal information on set data could help protect users' privacy.

## Acknowledgments and Disclosure of Funding

Tang and Yang's research is supported in part by the Hong Kong Research Grants Council under ECS grant 21214720, City University of Hong Kong under Project 9610465, and Alibaba Group through Alibaba Innovative Research (AIR) Program. Zhou's research is supported in part by City University of Hong Kong under Project 7200694. The authors thank the HKTVmall Open Databank for providing the HKTVMALL dataset. All opinions, findings, conclusions, and recommendations in this paper are those of the authors and do not necessarily reflect the views of the funding agencies.

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
