## B  Proof of Theorem 2

Without loss of generality, we assume that the probability of the empty set in $q(\cdot)$ is always 0. Let $V$ be the set of all items. We prove the theorem by an induction on $|V|$.

Basis Step: When $|V| = 1$, there is only one item in $V$ and suppose the item is $v_1$. Then $q(\{v_1\}) = 1$. We can set $p(v_1|\emptyset) = 1, p(\kappa|\{v_1\}) = 1$. Then $p(\{v_1\}) = p(v_1|\emptyset) \cdot p(\kappa|\{v_1\}) = 1$, which means $q = p$.

Inductive Step: Suppose the theorem holds for $|V| = m$. We discuss the case when $|V| = m + 1$.

Suppose $v_i$ is the $i$-th item. Let $V = V' \cup \{v_{m+1}\}$ and $V' = \{v_1, v_2, ..., v_m\}$. We define two sets $\mathcal{S}_{include} = \{S | v_{m+1} \in S\}$ and $\mathcal{S}_{exclude} = \{S | v_{m+1} \notin S\}$

First, we consider the conditional distribution $q_1$ on the condition that $v_{m+1} \notin S$. Then

$$q_1(S) = \frac{q(S)}{\sum_{A \in \mathcal{S}_{exclude}} q(A)}$$

Note that $q_1$ is a distribution over all subsets of $V'$ and $|V'| = m$. From the inductive hypothesis, we can build a distribution $p_1$ by setting up $p_1(\kappa|S)$ and $p_1(v|S)$ for $v \in V'$ and $S \in \mathcal{S}_{exclude}$, such that $p_1 = q_1$.

For $S \in \mathcal{S}_{exclude} \setminus \{\emptyset\}$ and $v \in V'$, we set

$$p(\kappa|S) = p_1(\kappa|S)$$
$$p(v|S) = p_1(v|S)$$

and

$$p(v|\emptyset) = p_1(v|\emptyset) \sum_{A \in \mathcal{S}_{exclude}} q(A)$$

Now, we show that, for any $S \in \mathcal{S}_{exclude}$, we have $p(S) = q(S)$.

$$p(S) = p_1(S) \times \sum_{A \in \mathcal{S}_{exclude}} q(A) = q_1(S) \times \sum_{A \in \mathcal{S}_{exclude}} q(A) = q(S)$$

Second, we consider $S \in \mathcal{S}_{include}$. We set

$$p(v_{m+1}|\emptyset) = \sum_{A \in \mathcal{S}_{include}} q(A)$$

$$p(\kappa|\{v_{m+1}\}) = \frac{q(\{v_{m+1}\})}{\sum_{A \in \mathcal{S}_{include}} q(A)}$$

For $S = \{v_{m+1}\}$, we have

$$p(\{v_{m+1}\}) = p(v_{m+1}|\emptyset) \times p(\kappa|\{v_{m+1}\}) = q(\{v_{m+1}\})$$

For $S \in \mathcal{S}_{include} \setminus \{\{v_{m+1}\}\}$, we can map it to $S \setminus \{v_{m+1}\}$. We define

$$\mathcal{S}'_{include} = \left\{ S \setminus \{v_{m+1}\} \middle| S \in \mathcal{S}_{include} \setminus \{\{v_{m+1}\}\} \right\}$$

The conditional distribution $q_2$ on the condition $v_{m+1} \in S$ is a distribution over $\mathcal{S}'_{include}$. Then, for any $S \in \mathcal{S}'_{include}$,

$$q_2(S) = \frac{q(S \cup \{v_{m+1}\})}{\sum_{A \in \mathcal{S}'_{include}} q(A \cup \{v_{m+1}\})}$$

Note that $q_2$ is a distribution over all subsets of $V'$ and $|V'| = m$. From the inductive hypothesis, we can build a distribution $p_2$ by setting up $p_2(\kappa|S)$ and $p_2(v|S)$ for $v \in V'$ and $S \in \mathcal{S}_{exclude}$, such that $p_2 = q_2$.

For $S \in \mathcal{S}_{include} \setminus \{\{v_{m+1}\}\}\}$, we set

$$p(\kappa|S) = p_2(\kappa|S \setminus \{v_{m+1}\})$$
$$p(v|S) = p_2(v|S \setminus \{v_{m+1}\})$$

and

$$p(v|\{v_{m+1}\}) = p_2(v|\emptyset) \times \sum_{A \in \mathcal{S}'_{include}} q(A \cup \{v_{m+1}\}).$$

For any $S \in \mathcal{S}_{include} \setminus \{\{v_{m+1}\}\}$, we have

$$p(S) = p_2(S \setminus \{v_{m+1}\}) \times \sum_{A \in \mathcal{S}'_{include}} q(A \cup \{v_{m+1}\})$$
$$= q_2(S \setminus \{v_{m+1}\}) \times \sum_{A \in \mathcal{S}'_{include}} q(A \cup \{v_{m+1}\})$$
$$= q(S)$$

Therefore, we have constructed $p(v|S)$ and $p(\kappa|S)$ such that $p(S) = q(S)$ for any $S \subseteq V$. The theorem holds when $|V| = m + 1$. This completes the induction.

## C   Optimal Size Distribution under $\ell_1$-Distance

In Section 5, we want to minimize the KL-divergence between $q(S)$ and $p^*(S)$. Here we discuss the situation when we try to minimize the $\ell_1$-distance. The constraints are the same as those in minimizing KL-divergence, while the objective becomes

$$\min_{q_1,\dots,q_K} \ell_1(q, p^*) = \sum_S ||q(S) - p^*(S)||_1$$

Then we define

$$f_k(q_k) = \sum_{|S|=k} ||q(S) - p^*(S)||_1$$

Similar to the discussion in Section 5, we check the derivative $f'_k(p^*_k)$,

$$f'_k(q_k) = \sum_{|S|=k} q^{(k)}(S)\text{sgn}[q_k q^{(k)}(S) - p^*_k p^{(k)*}(S)]$$

Here sgn[·] is the sign function[2]. By taking $q_k = p_k^*$, we have

$$f_k'(p_k^*) = \sum_{|S|=k} q^{(k)}(S)\text{sgn}[q^{(k)}(S) - p^{*(k)}(S)]$$

Therefore, probably $f_k'(p_k^*)$ varies for different $k$, if the distance between $q^{(k)}$ and $p^{*(k)}$ varies for different $k$. As a result, $q_k = p_k^*$ probably is not the optimal size distribution for minimizing the $\ell_1$-distance under Assumption 1. Moreover,

1. When $q^{(k)}$ and $p^{*(k)}$ are far away form each other. Then when $q^{(k)}(S)$ is large, $p^{*(k)}(S)$ is small. It will bring a large positive term to $f_k'(p_k^*)$. If $q^{(k)}(S)$ is small and $p^{*(k)}(S)$ is large, it will bring a small negative term. Thus $f_k'(p_k^*)$ is large when $q^{(k)}$ and $p^{*(k)}$ is far away form each other.

2. When $q^{(k)}$ and $p^{*(k)}$ are close, intuitively, $f_k'(p_k^*)$ should be close to 0.

Besides the above intuitive analysis, we also employ numerical simulations to explore the relationship between $f_k'(p_k^*) = \sum_{|S|=k} q^{(k)}(S)\text{sgn}[q^{(k)}(S) - p^{*(k)}(S)]$ and $\ell_1(q^{(k)}(S), p^{*(k)}(S))$. We randomly generate the two distributions $q^{(k)}$ and $p^{*(k)}$ as follows.

1. We treat $q^{(k)} = (q_1^{(k)}, q_2^{(k)}, \ldots, q_{1000}^{(k)})$ and $p^{*(k)} = (p_1^{*(k)}, p_2^{*(k)}, \ldots, p_{1000}^{*(k)})$ as two 1000-dimensional vectors. We randomly and independently generate each entry of $q^{(k)}$ and $p^{*(k)}$.

2. Entries of $q^{(k)}$ and $p^{*(k)}$ are both sampled from a uniform distribution $U[0,1]$. We also employ a standard Gaussian distribution $\mathcal{N}(0,1)$ to generate the random entries. After sampling all entries, we normalize both $q^{(k)}$ and $p^{*(k)}$ to make them probability distributions.

3. We generate 100,000 pairs of random $q^{(k)}$ and $p^{*(k)}$. For each pair, we calculate $\ell_1(q^{(k)}(S), p^{*(k)}(S))$ and $\sum_{i=1}^{1000} q_i^{(k)}\text{sgn}[q_i^{(k)} - p_i^{*(k)}]$. Then we plot all the 100,000 pairs in Fig. 4.

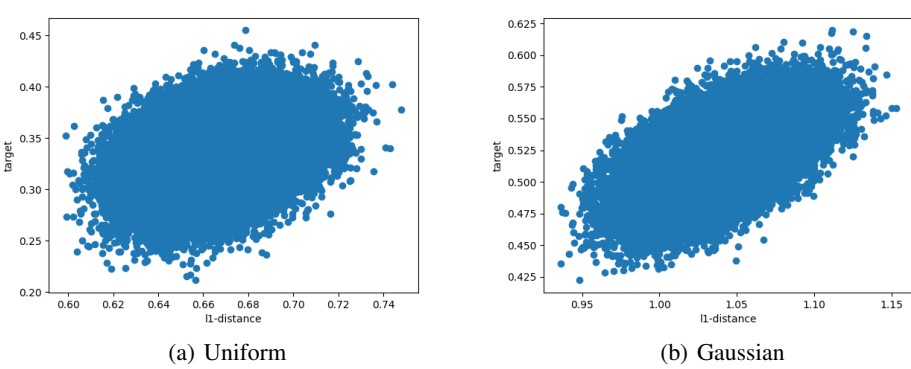

(a) Uniform                                    (b) Gaussian

Figure 4: $\ell_1$-distance $\ell_1(q^{(k)}(S), p^{*(k)}(S))$ v.s. $\sum_{i=1}^{1000} q_i^{(k)}\text{sgn}[q_i^{(k)} - p_i^{*(k)}]$. "target" indicates the value of $\sum_{i=1}^{1000} q_i^{(k)}\text{sgn}[q_i^{(k)} - p_i^{*(k)}]$.

Fig. 4 clearly shows the positive correlation between $\ell_1(q^{(k)}(S), p^{*(k)}(S))$ and value of Eq. (C) using $q^{(k)}$ and $p^{*(k)}$ as the inputs.

The above discussion and simulation results demonstrate that if $q^{(k)}$ is a bad estimation of $p^{(k)}$, we need to invest less on $k$. This is similar to the case of minimizing the KL-divergence discussed in Section 5.

---

[2]sgn[$x$] = 1 when $x > 0$; sgn[$x$] = 0 when $x = 0$; sgn[$x$] = −1 when $x < 0$.

# D   Generalization Ability of Sparse Item Graph

One may question if the sparse item graph constructed from training samples is too restricted such that many possible sets of items cannot be generated by it. We report the ratios of testing sets that cannot be generated by the sparse item graph in Table 5 and Table 6. It can be seen that on average only less than 2% testing sets cannot be generated by the sparse item graph built upon training samples. The reason may be that the number of possible pairwise co-occurrences of items is much less than $O(n^2)$, since many pairs of items are rarely purchased together if the two items have very different functions.

Table 5: Ratio of testing sets that cannot be generated by the sparse item graph constructed from training data on TMALL.

| Group | Order Size | | | | | Total Ratio |
|---|---|---|---|---|---|---|
| | 1 | 2 | 3 | 4 | $\geq 5$ | |
| 0 | 0.0000 | 0.0058 | 0.0036 | 0.0023 | 0.0038 | 0.0154 |
| 1 | 0.0000 | 0.0063 | 0.0036 | 0.0023 | 0.0038 | 0.0160 |
| 2 | 0.0000 | 0.0078 | 0.0043 | 0.0024 | 0.0034 | 0.0180 |
| 3 | 0.0000 | 0.0080 | 0.0045 | 0.0025 | 0.0041 | 0.0190 |
| 4 | 0.0000 | 0.0089 | 0.0047 | 0.0024 | 0.0038 | 0.0197 |
| 5 | 0.0000 | 0.0085 | 0.0046 | 0.0027 | 0.0040 | 0.0198 |
| 6 | 0.0000 | 0.0111 | 0.0059 | 0.0031 | 0.0039 | 0.0240 |
| 7 | 0.0000 | 0.0106 | 0.0056 | 0.0032 | 0.0041 | 0.0234 |
| mean | 0.0000 | 0.0084 | 0.0046 | 0.0026 | 0.0039 | 0.0194 |

Table 6: Ratio of testing sets that cannot be generated by the sparse item graph constructed from training data on HKTVMALL.

| Group | Order Size | | | | | Total Ratio |
|---|---|---|---|---|---|---|
| | 1 | 2 | 3 | 4 | $\geq 5$ | |
| 0 | 0.0000 | 0.0018 | 0.0005 | 0.0002 | 0.0002 | 0.0027 |
| 1 | 0.0000 | 0.0014 | 0.0004 | 0.0001 | 0.0002 | 0.0022 |
| 2 | 0.0000 | 0.0013 | 0.0004 | 0.0001 | 0.0002 | 0.0020 |
| 3 | 0.0000 | 0.0012 | 0.0003 | 0.0002 | 0.0002 | 0.0018 |
| 4 | 0.0000 | 0.0014 | 0.0004 | 0.0002 | 0.0002 | 0.0023 |
| 5 | 0.0000 | 0.0009 | 0.0003 | 0.0001 | 0.0002 | 0.0015 |
| 6 | 0.0000 | 0.0010 | 0.0003 | 0.0001 | 0.0002 | 0.0016 |
| 7 | 0.0000 | 0.0013 | 0.0004 | 0.0001 | 0.0002 | 0.0020 |
| mean | 0.0000 | 0.0013 | 0.0004 | 0.0002 | 0.0002 | 0.0020 |

# E   Size-wise Analysis

We report more detailed experimental results in this section. The results can help readers better understand the advantages of our model and the effect of our size-bias trick. Moreover, based on the results, we will show that we can even combine Histogram with our models to build even stronger models for learning set distributions.

Table 7: The average size distribution on TMALL

| Method | Order Size | | | | |
|---|---|---|---|---|---|
| | 1 | 2 | 3 | 4 | $\geq 5$ |
| Size-bais | 0.5015 | 0.2608 | 0.1577 | 0.0654 | 0.0146 |
| Benchmark | 0.3844 | 0.1780 | 0.1363 | 0.1033 | 0.1980 |
| GRU2Set | 0.3755 | 0.1916 | 0.1394 | 0.0999 | 0.1937 |
| SetNN | 0.3815 | 0.2097 | 0.1366 | 0.0913 | 0.1809 |
| MRW | 0.3525 | 0.1966 | 0.1392 | 0.0983 | 0.2134 |
| DCM | 0.3844 | 0.1780 | 0.1363 | 0.1033 | 0.1980 |

Table 8: The average size distribution on HKTVMALL

| Method | Order Size | | | | |
|---|---|---|---|---|---|
| | 1 | 2 | 3 | 4 | ≥5 |
| Size-bais | 0.4947 | 0.2228 | 0.1424 | 0.0904 | 0.0498 |
| Benchmark | 0.4155 | 0.1680 | 0.1088 | 0.0819 | 0.2258 |
| GRU2Set | 0.4001 | 0.1702 | 0.1092 | 0.0813 | 0.2392 |
| SetNN | 0.4048 | 0.1800 | 0.1155 | 0.0814 | 0.2183 |
| MRW | 0.4004 | 0.1938 | 0.1321 | 0.0912 | 0.1826 |
| DCM | 0.4155 | 0.1680 | 0.1088 | 0.0819 | 0.2258 |

We first report the average size distribution of each method as well as the size distribution calibrated by our size-bias trick in Table 7 and Table 8. We can see that before applying the size-bias trick, all methods learn similar size distributions which are very close to the size distribution of the training data (Histogram). The effect of applying our size-bias trick is to increase the ratio of small-sized orders and reduce the number of large-sized orders in our prediction $\mathcal{S}_{pred}$.

Table 9: The average size-wise Overlap on TMALL

| Method | Order Size | | | | | Total Overlap |
|---|---|---|---|---|---|---|
| | 1 | 2 | 3 | 4 | ≥5 | |
| Benchmark | 0.3692 | 0.1166 | 0.0325 | 0.0056 | 0.0010 | 0.5249 |
| +Size Bias | 0.3797 | 0.1267 | 0.0337 | 0.0049 | 0.0004 | 0.5454 |
| GRU2Set | 0.3494 | 0.1185 | 0.0407 | 0.0076 | 0.0010 | 0.5172 |
| +Size Bias | 0.3764 | 0.1359 | 0.0449 | 0.0053 | 0.0002 | 0.5627 |
| SetNN | 0.3475 | 0.1247 | 0.0407 | 0.0070 | 0.0008 | 0.5207 |
| +Size Bias | 0.3751 | 0.1349 | 0.0448 | 0.0052 | 0.0002 | 0.5602 |
| MRW | 0.3334 | 0.1129 | 0.0366 | 0.0067 | 0.0008 | 0.4904 |
| +Size Bias | 0.3682 | 0.1253 | 0.0398 | 0.0046 | 0.0002 | 0.5381 |
| DCM | 0.3691 | 0.1033 | 0.0242 | 0.0025 | 0.0004 | 0.4995 |
| +Size Bias | 0.3797 | 0.1198 | 0.0266 | 0.0018 | 0.0001 | 0.528 |
| hybrid(Benchmark + GRU2Set) | 0.3797 | 0.1359 | 0.0449 | 0.0053 | 0.0002 | **0.5660** |
| hybrid(Benchmark + SetNN) | 0.3797 | 0.1349 | 0.0448 | 0.0052 | 0.0002 | 0.5648 |

Table 10: The average size-wise Overlap on HKTVMALL

| Method | Order Size | | | | | Total Overlap |
|---|---|---|---|---|---|---|
| | 1 | 2 | 3 | 4 | ≥5 | |
| Benchmark | 0.4080 | 0.1271 | 0.0365 | 0.0103 | 0.0029 | 0.5848 |
| +Size Bias | 0.4142 | 0.1349 | 0.0388 | 0.0106 | 0.0021 | 0.6006 |
| GRU2Set | 0.3788 | 0.1253 | 0.0409 | 0.0116 | 0.0032 | 0.5598 |
| +Size Bias | 0.4091 | 0.1388 | 0.0470 | 0.0125 | 0.0023 | 0.6097 |
| SetNN | 0.3792 | 0.1268 | 0.0414 | 0.0111 | 0.0030 | 0.5615 |
| +Size Bias | 0.4064 | 0.1370 | 0.0461 | 0.0119 | 0.0022 | 0.6036 |
| MRW | 0.3836 | 0.1248 | 0.0403 | 0.0107 | 0.0028 | 0.5622 |
| +Size Bias | 0.4041 | 0.1319 | 0.0419 | 0.0107 | 0.0020 | 0.5906 |
| DCM | 0.4079 | 0.1225 | 0.0316 | 0.0074 | 0.0021 | 0.5715 |
| +Size Bias | 0.4112 | 0.1283 | 0.0338 | 0.0075 | 0.0019 | 0.5827 |
| hybrid(Benchmark + GRU2Set) | 0.4142 | 0.1388 | 0.0470 | 0.0125 | 0.0023 | **0.6148** |
| hybrid(Benchmark + SetNN) | 0.4142 | 0.1370 | 0.0461 | 0.0119 | 0.0022 | 0.6114 |

We then report the effectiveness of each method on orders of different sizes in Table 9 and Table 10. To better understand the result, we use the metric $Overlap$ [Wu et al., 2021] which can be converted

to $\ell_1$-distance since

$$Overlap(\mathcal{S}_{test}, \mathcal{S}_{pred}) = \sum_{S \in \mathcal{S}_{test} \cup \mathcal{S}_{pred}} \min\{\frac{N_{test}(S)}{|\mathcal{S}_{test}|}, \frac{N_{pred}(S)}{|\mathcal{S}_{pred}|}\} = (1 - \ell_1(\mathcal{S}_{test}, \mathcal{S}_{pred})) * 2$$

We decompose $Overlap(\mathcal{S}_{test}, \mathcal{S}_{pred})$ as

$$Overlap(\mathcal{S}_{test}, \mathcal{S}_{pred}) = o_1 + o_2 + o_3 + o_4 + o_5,$$

where $o_k = \sum_{|S|=k} \min\{\frac{N_{test}(S)}{|\mathcal{S}_{test}|}, \frac{N_{pred}(S)}{|\mathcal{S}_{pred}|}\}$ for $k = 1, 2, 3, 4,$ and $o_5 = \sum_{|S| \geq 5} \min\{\frac{N_{test}(S)}{|\mathcal{S}_{test}|}, \frac{N_{pred}(S)}{|\mathcal{S}_{pred}|}\}$.

Using the results in Table 9 and Table 10, as well as the size distributions reported in Table 7 and Table 8, we can calculate the "cost-effectiveness" of guessing size-$k$ sets in $\mathcal{S}_{pred}$, which is the ratio between the overlap on size-$k$ sets and the ratio of size-$k$ sets in $\mathcal{S}_{pred}$. By simple calculation we find that the cost-effectiveness of small sets is often greater than that of large sets. This is not surprising as in both TMALL and HKTVMALL small sets are more abundant than large sets. In addition, the number of possible small sets is much less than the number of possible large sets due to the combinatorial explosion of sets. As a result, it is easier to learn the probabilities of small sets than large sets, which is consistent to our Assumption 1. Therefore, if we increase the number of small sets in $\mathcal{S}_{pred}$ and reduce the number of large sets in $\mathcal{S}_{pred}$, the marginal gain on the overlap on small sets has a good chance to be greater than our loss on the overlap on large sets. This motivates us to apply our size-bias trick.

Another important finding from Table 9 and Table 10 is that Histogram's $o_1$ is often the best while our models' $o_2$, $o_3$ and $o_4$ are better than those of Histogram. We explain this from a perspective of the effective sample size and the number of possible size-$k$ sets.

Take TMALL as an example. In our training data, on average we have $100,000 * 38.44\% = 38,440$ size-1 orders. Compared to the number of items (1,363), 38,440 training samples are enough for us to learn the set distribution for size-1 sets well. Moreover, the distribution of size-1 sets actually is a standard discrete distribution. Kamath et al. [2015] proves that for learning a discrete distribution, Histogram can achieve the min-max rate and is the asymptotically optimal method. Therefore, Histogram often achieves the best overlap on size-1 sets.

However, when it comes to sets of multiple items, the number of training samples is not enough compared to the number of possible size-$k$ sets for $k > 1$. Moreover, the distribution of sets of multiple items is more than a standard discrete distribution, as similar sets may have some connections to each other. Such information is totally ignored by Histogram but well captured by our models. Note that Histogram has no extrapolation ability as it cannot model the probability of any unseen set. Therefore, it is not surprising that our models have better overlap on size-2, size-3 and size-4 sets than Histogram.

The baseline methods MRW [Wu et al., 2021] and DCM [Benson et al., 2018] both try to catch the connections between similar sets but their performance is not good. A possible reason is that both MRW and DCM make too strong assumptions on how random sets are generated, while our models adopt deep neural nets whose expressive power is extremely strong.

**Hybrid Model** One interesting finding based on the detailed experimental results is that we can even combine Histogram with our models to build stronger models for learning set distributions. Specifically, for TMALL and HKTVMALL, we use Histogram to construct the probabilities of size-1 sets and use our models for sets of multiple items. We call such a model a Hybrid model and we report its performance in Table 9 and Table 10. Since the hybrid model has the advantages of both Histogram and our models, it achieves the best performance in Table 9 and Table 10.