# OpenReview forum: "Sequence-to-Set Generative Models"
_NeurIPS.cc/2022/Conference — NeurIPS 2022 Accept_

### Official Review · Reviewer_DMSE · 2022-07-10

**Rating:** 6
**Confidence:** 1
**Soundness:** 3 good
**Presentation:** 3 good
**Contribution:** 3 good

**Summary:**

This paper tackles the challenge of developing generative models for sets, which have the property of permutation invariance. The authors first present a method for converting a sequence generative model to a set generative model based on maximum likelihood.  Additionally, they also propose an importance sampling algorithm to estimate the gradient of the log-likelihood function to improve the efficiency.  The authors further propose two parameterizations of the sequence-to-set model, which are GRU2Set and SetNN.  Finally, the authors propose a size-bias trick that can help reduce the distance between the ground truth set distribution and the learned distribution.

Empirical evaluations have been performed on two real-world datasets to convey the effectiveness of their method.

**Questions:**

1. The formats of the citations are incorrect in a few places including the very first sentence in the introduction, making the text hard to read.
2. What are the authors boldfacing in the tables?
3. It’s a bit hard for me to understand why the performance for the > 7 group is even better than the smaller sizes.  Learning the set distribution for more items is an easier problem?  What’s the intuition here.

**Limitations:**

No ethical concerns.  Other limitations discussed in earlier sections.

**Strengths And Weaknesses:**

Strengths:
1. The paper studies an important and understudied problem and proposed an interesting method
3. The empirical results are strong against baseline methods

Weaknesses:
1. The presentation of the paper is a bit hard to follow.  I would suggest the authors to illustrate the size-bias trick with a running example.

---

> ### Author Response · Authors · 2022-08-02
> **Incorrect citation formats**
>
> Thank you for pointing out the incorrect formats. We have fixed the issue in the revised paper.

---

> ### Author Response · Authors · 2022-08-02
> **Boldfacing numbers in the tables**
>
> The boldfacing numbers in each column are the top-2 results among all methods on each group of data.

---

> ### Author Response · Authors · 2022-08-02
> **Why the performance for the > 7 group is even better than the smaller sizes?**
>
> We guess there might be some misunderstanding on the meaning of ``group''. Since the purchasing behaviors of consumers evolve over time [Wu et al., 2021], we split both TMALL and HKTVMALL into groups where each group contains orders in a specific month. The group numbers are irrelevant to the sizes of orders.

---

> ### Author Response · Authors · 2022-08-02
> **Running example of size-bias trick**
>
> Please refer to our response ``Analysis on size-wise variation'' to Reviewer 1KAT.

---

### Official Review · Reviewer_uw8U · 2022-07-11

**Rating:** 6
**Confidence:** 3
**Soundness:** 3 good
**Presentation:** 3 good
**Contribution:** 3 good

**Summary:**

The authors in this paper propose Sequence-to-Set that aims to convert any likelihood based sequence generative model to a set generative model. The proposed model first introduces a probability (likelihood) over a set $S$ induced from the generating path of the sequence generative model. Then an efficient importance sampling method is devised to tackle the training challenge of maximizing the set likelihood. The authors also present two instance of the proposed model, GRU2Set and SetNN to capture the complex relations among data points. Besides, a size-bias trick is designed to learn better set distribution from data. The proposed model is evaluated on two e-commerce order datasets, and the results show that the proposed model outperforms its baselines.

**Questions:**

(1) I don't quite understand the definition of the indicator function $\tau$, is there any minor in $s_i=s \land a_i = a$ ?


**Strengths And Weaknesses:**

Strengths:

(1) The paper clearly identifies the problems with previous works and presents the idea of the proposed method.

(2) The paper is well-written and easy to follow.

(3) The paper provides a strong baseline for the set generative model, and several attempts can be tried base on the proposed method.

(4) The experiments are well-designed.


Weaknesses:

(1) More deeper analysis about the experiments. for example, without Size-bias, the proposed model has different performance on the two datasets. The authors should give more discussion.

(2) Empirically, the Size-Bias trick is the key for the proposed method and it can be used as a plug-and-play module to improve any generative model. It seems that the Size-Bias trick improves the proposed two models larger than baselines. The authors should give detailed explanation.

---

> ### Author Response · Authors · 2022-08-02
> **Different performance on the two datasets**
>
> Although both TMALL and HKTVMALL are customer order datasets, the two datasets are about two different populations of consumers. TMALL contains orders purchased by users in one city in mainland China [Wu et al., 2021], while HKTVMALL's orders were made by users in Hong Kong. We believe that the different performance on the two datasets is mainly caused by the different consumer behaviors in the two datasets.

---

> ### Author Response · Authors · 2022-08-02
> **About the indicator function**
>
> We explain it with an example. Suppose we have one training sample $S=${1,2,4} and we have already visited item $4$. Assume that item $4$ is connected to item 1, item 2 and item 3. Then we have the following 4 actions,
>
> $a_1$: choosing item $1$
>
> $a_2$: choosing item $2$
>
> $a_3$: choosing item $3$
>
> $a_{\kappa}$: choosing the stop item $\kappa$
>
> By using the set of all items visited so far as the state, we have
>
> $\tau(a_1$, {4}, {1,2,4}$)$ = $\tau(a_2$, {4}, {1,2,4}$)$ = 1 and
>
> $\tau(a_3$, {4}, {1,2,4}$)$ = $\tau(a_{\kappa}$, {4}, {1,2,4}$)$ = 0.
>
> This is because if we choose item 3 or the stop item $\kappa$, then we can never output the set {1,2,4}.

---

> ### Author Response · Authors · 2022-08-02
> **Detailed explanation of the effect of size-bias trick**
>
> Please refer to our response ``Analysis on size-wise variation'' to Reviewer 1KAT.

---

### Official Review · Reviewer_1KAT · 2022-07-11

**Rating:** 6
**Confidence:** 4
**Soundness:** 3 good
**Presentation:** 3 good
**Contribution:** 3 good

**Summary:**

This paper focuses on the problem of modeling sets via sequence models via generative approaches. Mainly this paper considers all the possible state-action paths that would yield the same set and approximately marginalizes over them for MLE estimation. An approach similar to the popular "REINFORCE"/score-matching trick is proposed to define a Monte-Carlo based estimator for the gradient of the problem and importance sampling is done to efficient estimate the gradient for training. Two instantiations of the model are proposed -- GRU2set and SetNN. The first uses an autoregressive decoder's states to represent items in the sets, the second approach defines an equivariant MLP based function to represent items generated so far in the set. Size-bias trick is also introduced which biases the learning of set generation according to the size. The experiments are done on ecommerce order datasets, and compared with histogram based baselines and other relevant models

**Questions:**

How does “sparse item graph" constructed form training data generalize to test sets?
Confused about their definition of "induce"



**Strengths And Weaknesses:**

This paper tackles an important problem which would be of great interest to retro fit essentially sequnce-based architecture for modelling sets. The main approach is clearly explained and is sound.

My main concern is that the empirical results show that the simple histogram-based baseline outperforms most of the models. The proposed approach trails until it applies the size bias trick to see modest gains over the baseline.

Analysis of performance variation according to the size-wise variation of the set would improve the paper.

Why not use transformers? A simple transformer model can come close to modelling set properties by disabling positional embeddings. Comparison to transformers/SetTransofrmers is crucial given their ubiquity.

Related work: multilabel classification literature faces the same issues, a discussion about the approaches in this paradigm would be useful. For example: "Orderless Recurrent Models for Multi-label Classification", Yazici et al.

The paper is awkwardly written in parts – the size bias trick is very difficult to understand.

Line 235/236 is very confusing

---

> ### Author Response · Authors · 2022-08-01
> **Why is Histogram so strong?**
>
> A set distribution is a special kind of discrete distribution. Histogram is a very strong method for learning discrete distributions. Let $d$ be the size of the support set of the ground truth discrete distribution and $m$ be the number of training samples. Histogram can guarantee an $O((d/m)^{1/2})$ $\ell_1$-distance and a min-max rate on discrete distribution is $\Omega((d/m)^{1/2})$ (``On Learning Distributions from their Samples'', 2015).
>
> However, for set distribution learning, Histogram may not be optimal as similar sets may have some connections to each other. Histogram cannot capture such connections. Both our models and baseline models (DCM and MRW) try to capture such connections. The reason that our models outperform both DCM and MRW may be that we use deep neural nets whose expressive power is much stronger than that of DCM and MRW. The reason that Hisotgram is better than DCM and MRW may be that the number of training samples in our experiments is not small, especially for size-1 orders (single items). A detailed discussion about this issue can be found in our ``Analysis on size-wise variation''.
>
> We also want to emphasize that although Histogram is strong w.r.t. the $\ell_1$-distance metric, it has many obvious drawbacks. For example, Histogram has no ability for extrapolation since it can only model probabilities of sets appearing in the training dataset. Moreover, unlike our model where we also produce item embeddings and set embeddings which may be helpful in understanding the structure of the set distribution and downstream tasks, Histogram cannot provide such information.

---

> ### Author Response · Authors · 2022-08-01
> **Why not use transformers**
>
> Thank you for your advice. When we did the experiments, considering the computation costs and clarity, we tried to use some simple nets to show the effectiveness of our method. Compared to transformers with many multi-head attention blocks, the structure of GRU2Set or SetNN only has one latent layer and our embedding dimension is only 10. Based on our current experimental results, we can see that it is promising to use our method to convert a sequence model to a set model. For future work, we will try more advanced sequence models, for example, replacing GRU in GRU2Set with a transformer and using SetTransofrmer to replace MLP in SetNN.

---

> ### Author Response · Authors · 2022-08-01
> **Concerns about the sparse item graph**
>
> The sparse item graph models the pairwise co-occurrences of items. Note that the number of such co-occurrences (edges) is at most $O(n^2)$ where $n$ is the number of items. Intuitively, some pairs of items are rarely purchased together if the two items have very different functions. As a result, the number of possible edges may be much less than $O(n^2)$. Therefore, although the training samples are far from enough to have all possible sets, they may be enough to have the vast majority of the pairwise co-occurrences of items. In fact, for both TMALL and HKTVMALL, there are less than 2% testing orders that cannot be generated by the sparse item graph constructed from the training data.
>
> Moreover, the sparse item graph restricts the possible combinations of items, which plays a role similar to a regularizer and can help bring a better generalization ability.
>
> We say that a generating path can ``induce'' a set $S$ if the state-associated set (line 107) of the final state of the path is $S$.

---

> ### Author Response · Authors · 2022-08-02
> **Analysis on size-wise variation**
>
> Thank you for suggesting an analysis on size-wise variation. We conduct such an analysis by checking some of the intermediate results of our experiments. We use the experimental result on the first group of the TMALL dataset to show how the size-bias trick improves the quality of the learned set distribution. Moreover, we analyze why Histogram is a strong benchmark method.
>
>
> Let $N$ be the number of predicted orders and $M$ be the number of testing orders. $f(S)$ and $g(S)$ denote \#appearance of $S$ in predicted orders and testing orders, respectively. We use overlap [Wu et al., 2021] to show the performance.
>
> $overlap(predict, test)=\sum_{S} \min(f(S)/N , g(S)/M)$
>
> The larger the overlap, the smaller the $\ell_1$-distance, since
>
> $\ell_1(predict, test) = (1 - overlap(predict, test)) * 2$ [Wu et al., 2021].
>
> We decompose overlap into 5 parts by size as follows.
> For k=1,2,3,4, $o_k = \sum_{|S|=k} \min(f(S)/N , g(S)/M)$, and
> $o_5 = \sum_{|S|>=5} \min(f(S)/N , g(S)/M)$.
>
> Then
> $overlap(predict, test) = o_1 + o_2 + o_3 + o_4 + o_5$.
>
> Below, we use the first group of TMALL dataset as a running exmaple to show how the size-bias trick helps improve the quality of the learned set distribution.
>
> The size distribution of training data is
>
> [size=1, size=2, size=3, size=4, size$\ge$5]
>
> [0.305,   0.162,   0.144,   0.122,   0.267].
>
> The size distribution of predicted orders by GRU2Set trained from data is close to the set distribution of the training data,
>
> [size=1, size=2, size=3, size=4, size$\ge$5]
>
> [0.287,   0.195,   0.156,   0.118,   0.245].
>
> After applying the size-bias trick, the size distribution becomes
>
> [size=1, size=2, size=3, size=4, size$\ge$5]
>
> [0.422,   0.270,   0.191,   0.092,   0.024].
>
> $overlap(predict, test)        = o_1    + o_2    + o_3    + o_4    + o_5$
>
> histogram: overlap             = 0.2916 + 0.1095 + 0.0369 + 0.0069 + 0.0012 = 0.4462
>
> histogram + size bias: overlap = 0.2995 + 0.1213 + 0.0397 + 0.0065 + 0.0006 = 0.4676
>
> GRU2Set: overlap               = 0.2689 + 0.1179 + 0.0465 + 0.0086 + 0.0010 = 0.4428
>
> GRU2Set   + size bias: overlap = 0.2990 + 0.1310 + 0.0535 + 0.0069 + 0.0003 = 0.4907
>
>
> We make the following analysis of the experimental results.
>
> (i). Why does size-bias work?
>
> First, based on the definition of $o_k=\sum_{|S|=k} \min(f(S)/N , g(S)/M)$, increasing the number of size-$k$ predicted orders can help increase $o_k$. For example, after applying size-bias trick, the ratios of size-1, size-2 and size-3 orders are increased and $o_1$, $o_2$ and $o_3$ are all increased.
>
> On the other hand, if we decrease the number of size-$k$ orders in our prediction for large $k$, such as $k=4$ and $k\ge 5$, $o_k$ probably will be decreased. However, originally the overlap on orders of large size is not that good. Therefore, the negative effect on decreasing the ratio of large orders can be smaller than the advantages caused by increasing the ratio of small orders.
>
> (ii). Why histogram's $o_1$ is better and why our models' $o_2$, $o_3$, $o_4$ are better?
>
> Note that if we only care about size-1 orders, then the set distribution degenerates to a standard discrete distribution. In such a case, if the number of training samples is enough, Histogram can learn the distribution pretty well as it can theoretically guarantee an $O((d/m)^{1/2})$ $\ell_1$-distance. In our experiments, taking TMALL as an example, the number of size-1 orders in the training set is roughly $100,000*30.5$% $\gg d=n=1363$. Therefore, it is not surprising that Histogram performs the best w.r.t. $o_1$.
>
> When it comes to orders of multiple items, the training samples are usually not enough for Histogram to learn the distribution well. For example, in TMALL, on average we have $100,000*16.2$% size-2 training orders. Such a number probably may even be smaller than the number of all possible size-2 orders. One can easily find that the data sparsity issue for orders of bigger sizes is even more severe, which makes the convergence rate $O((d/m)^{1/2})$ of Histogram meaningless. In such a case, it is very important for a model to have a good extrapolation ability, which is the key that our models outperform Histogram on $o_2$, $o_3$, and $o_4$. Note that Histogram has no extrapolation ability as it cannot model the probability of any unseen set.
>
> An extra finding of our analysis is that we can build an even stronger model by combining Histogram with our model. The idea is to use Histogram to construct the probabilities of size-1 sets and use our model for orders of multiple items. We report this in the supplementary material of the revised paper.

---

> > ### Comment · Reviewer_1KAT · 2022-08-10
> > **Thanks for your detailed response**
> >
> > I would like to thank the authors for their response to my review (and apologize for my late response to the authors) and specifically I found the size-wise variation analysis and the discussion about histogram's performance helpful to better understand the effect of the proposed approach. I like the idea of combining histogram distribution for singletons and using the model for other set sizes. It seems the size-bias trick improves the proposed approach's performance but it diminishes the contribution of larger sized sets at the same time. The response clarified that this works out well for the datasets considered because the overlap at higher sizes is low. However, I still wonder what the effect of this trick would be in general. For example, how would the proposed approach fair in settings where larger sparser sets comprise a heavy tail?

---

> > > ### Author Response · Authors · 2022-08-10
> > > **Author-Reviewer Discussion Ends**
> > >
> > > Thank you for your response. Unfortunately, we received an email saying that we are not allowed to answer reviewers' questions now. We would be happy to reply to your new comment if the PC chairs, the senior AC, and the AC allow us to do so.

---

> ### Author Response · Authors · 2022-08-09
> **Related work and Line 235/236**
>
> Thanks for suggesting the multi-label classification paper. In our revised paper, we review this paper as well as another recent study in this line in Related Work.
>
> We also rephrase the original line 235/236. Now it is line 240/241. Basically, the equation at the bottom of page 6 is to decompose the process of generating a random set into two steps. The first step is to randomly decide $k$, the size of the set, and the second step is to generate a random size-$k$ set. The rationale is that a set distribution is a mixture of $K$ sub-models, where $K$ is the largest possible set size and each sub-model is a distribution over all sets of a specific size.

---

### Author Response · Authors · 2022-08-02
**Response to all reviewers**

We thank the reviewers for their time and efforts, and for providing valuable feedback in their detailed reports. Below, we reply to the concerns of the reviewers. We also take some of the suggestions of the reviewers and revise our paper accordingly. Specifically, in the revision, we fix the format issue of citations, add an analysis of the generalization ability of sparse item graph and a size-wise analysis to Appendix, which can help readers better understand the advantages of our model and the effect of our size-bias trick. Moreover, in the size-wise analysis, we find that a hybrid model combining both Histogram and our models can achieve even better performance.

---

### Meta-Review · Area_Chair_DJu7 · 2022-08-28

**Recommendation:** Accept
**Confidence:** Certain

**Metareview:**

Reviewers found the problem to be well-motivated and the results convincing. They objected to not using transformers and some difficulties to outperform the strong histogram baseline and presentation of the key size-bias trick.

The authors addressed some of the issues raised and all reviewers rated weak accept.

**Award:**

No

---

### Decision · Program_Chairs · 2022-09-14

Accept